# Isotopic Niche of Syntopic Granivores in Commercial Orchards and Meadows

**DOI:** 10.3390/ani11082375

**Published:** 2021-08-11

**Authors:** Linas Balčiauskas, Raminta Skipitytė, Andrius Garbaras, Vitalijus Stirkė, Laima Balčiauskienė, Vidmantas Remeikis

**Affiliations:** 1Nature Research Centre, Akademijos 2, 08412 Vilnius, Lithuania; raminta.skipityte@ftmc.lt (R.S.); vitalijus.stirke@gamtc.lt (V.S.); laima.balciauskiene@gamtc.lt (L.B.); 2Center for Physical Sciences and Technology, Saulėtekio av. 3, 02300 Vilnius, Lithuania; andrius.garbaras@ftmc.lt (A.G.); vidmantas.remeikis@ftmc.lt (V.R.)

**Keywords:** *Apodemus*, *Micromys*, inter and intraspecific competition, trophic niche, carbon-13 and nitrogen-15 isotopes, commercial orchards and berry plantations

## Abstract

**Simple Summary:**

Granivorous murids, namely striped field (*Apodemus agrarius*), yellow-necked (*Apodemus flavicollis*), and harvest (*Micromys minutus*) mice, occur in a variety of habitats and live syntopically in agricultural areas. Agroecosystems may be quite complex isotopically with *δ*^15^N values being influenced by many internal and external fluxes. Using isotopic (*δ*^15^N and *δ*^13^C) compositions from hair samples, we analysed isotopic niches of granivores in apple and plum orchards, raspberry and currant plantations, and nearby meadows in Lithuania. As the main hypothesis, we expected differences in the isotopic niches of these species (being a proxy for their diet), minimising interspecific competition. Striped field and yellow-necked mice were trapped in every habitat. Therefore, syntopic co-occurrence of granivores depended on the presence of harvest mice in the apple orchards, raspberry plantations, and meadows that served as control habitats. All species were fully separated according to *δ*^15^N values, presuming different amounts of food of animal origin in their diet. The separation of species according to *δ*^13^C was not expressed in all habitats. The core dietary niches of these species were fully separated in the apple orchards and raspberry plantations. Intraspecific differences of the isotopic niche were not present in any of the three species: that is, resources were equally used by males and females, adults, subadults, and juveniles.

**Abstract:**

In agricultural habitats, diets and trophic positions of syntopic granivorous small mammals are not known sufficiently. Agroecosystems may be quite complex isotopically and the most complex situation concerns the nitrogen-15 isotope as *δ*^15^N values are influenced by many internal and external fluxes. We analysed the isotopic niches of striped field (*Apodemus agrarius*), yellow-necked (*Apodemus flavicollis*), and harvest (*Micromys minutus*) mice living sympatrically and syntopically in apple and plum orchards, raspberry and currant plantations, and nearby meadows that were used as control habitats. Carbon (*δ*^13^C) and nitrogen (*δ*^15^N) stable isotope ratios from hair samples were used as a proxy for their diet. As the main hypothesis, we expected differences in the isotopic niches of these three species, minimising interspecific competition. All species were fully separated according to *δ*^15^N values, presuming different amounts of food of animal origin in their diet. The separation of species according to *δ*^13^C was not expressed in all habitats. The core dietary niches of these species were fully separated in the apple orchards and raspberry plantations. Intraspecific differences of the isotopic niche were not present in any of the three species: that is, resources were equally used by males and females, adults, subadults, and juveniles.

## 1. Introduction

Three granivorous rodents, namely the striped field mouse *Apodemus agrarius* (Pallas, 1771), the yellow-necked *mouse Apodemus flavicollis* (Melchior, 1834), and the harvest mouse *Micromys minutus* (Pallas, 1771), all occur in a wide variety of habitats but favour areas of tall, dense vegetation [1,2,3,4]. These closely related species are common in Central and Eastern Europe [5,6,7,8]. In Lithuania, populations of the above mentioned granivores may occur syntopically in moist habitats [9,10] but little is known about their diet and interspecific relationships.

Diets link animals with their environment and have an influence on physiological, behavioural, and morphological traits [11]. In the temperate climate zone, most rodent species belong to three groups according to their diet: granivores (*Apodemus* and *Micromys* species), herbivores (*Microtus* and *Arvicola* species), and omnivores (*Clethrionomys* and *Mus* species) [12,13]. Pineda-Munoz and Alroy in 2014 identified granivory as a readily identifiable diet type in mammals [14]. The ecological or evolutionary relevance of granivory is evidenced by the fact that several rodent species have modified incisors for shaving and husking seeds [15]. These structures evolved in species in the families of Cricetidae and Muridae [16].

Small mammals are not only part of food chains in agroecosystems but also are pests of many crops [17]. Their negative effects are not limited to their damage of agricultural crops but rather expand to their role in the distribution of weed seeds and the fact that they are carriers of various pathogens [18]. Granivorous animals play a dual role in plant regeneration [19], both enhancing it by dispersing seeds and reducing it by often consuming large quantities of seeds [20]. Although granivorous rodents are known for hoarding seeds, their ability to consume foods other than seeds allows them to survive variation in resource availability [21]. As a result, the quantity of seeds consumed by granivores may depend critically on the presence of other nearby resources in the environment that serve to attract or distract foraging granivores [22].

In commercial orchards and berry plantations in Lithuania, the dominant species is the common vole (*Microtus arvalis*), while *A. agrarius* and *A. flavicollis* are subdominants, although they dominate in some cases [23]. *M. minutus* is a rare species in commercial gardens, represented by just a few individuals [10].

With high dietary and microhabitat overlaps, some small mammal species could coexist primarily only due to passive, mutually avoiding and amicable interactions [24]. Another opportunity for coexistence is niche partitioning [25], which may be based on time [26,27], space or habitats [28,29], or resources [30,31].

Resource use or diet is one of the main dimensions of a niche [32]. An assessment of resource use may concern consumed food items (relevance or frequency) using methods of analysis such as assessing stomach contents, scats, cheek pouches, food caches, etc. [33]. In recent years, diet has been analysed by modern tools such as DNA, fatty acid, or stable isotope analyses [34,35,36,37]. We used stable carbon and nitrogen (*δ*^13^C and *δ*^15^N) isotope analysis as a reliable tool to investigate the trophic ecology of vertebrate animals including various rodent species [9,23,30,36,37,38,39].

Our study concentrated on a stable isotope-based evaluation of the trophic niches of three granivores species, namely striped field mouse, yellow-necked mouse, and harvest mouse, these being sympatric and syntopic simultaneously in commercial orchards, berry plantations, and nearby meadows, which are defined as control habitats. As a main hypothesis, we expected differences in their diet (expressed as a proxy by the isotopic space of the species) and that these differences minimise interspecific competition for food resources of the co-occurring species. With more resources available to granivores in this habitat, we expected an overlap of the isotopic niches in the meadows but not in the fruit orchards or berry plantations, as these are resource-scarce. We expected that intraspecific (age or gender based) differences will be present in all investigated habitats as a result of the competition for resources.

## 2. Materials and Methods

### 2.1. Study Site

Small mammals were trapped in 2018–2020 at 18 sites in Lithuania. At every site, an agricultural habitat (apple or plum orchard, or currant, raspberry, or highbush blueberry plantation) and a neighbouring control meadow or forest (not subjected to agricultural activity) were investigated (Figure 1a). Commercial habitats were of different age (young, medium-aged, and old) and with different intensities of agricultural measures in their maintenance (intensity defined as low, medium, or high). Additional details on the study sites and habitats are presented in previous publications [10,31].

### 2.2. Small Mammal Trapping and Sample Size

Small mammals were snap-trapped two times per year in summer and autumn, covering one or two agricultural habitats and a control habitat at every investigation site (see Figure 1). We used a trap line method, setting 25–100 traps at 5 m intervals per habitat and checking these once per day in the morning [40]. Total trapping effort was 25,503 trap days. The distribution of the trapping effort according to the habitats is shown in Table 1. In total, 168 trapping sessions (three-day long trapping per habitat) were conducted. In three of these sessions, in the high blueberry plantation, no small mammals were trapped. Therefore, this habitat was not analysed.

Mice were identified visually when checking the traps and stored in separate plastic bags. Animal age and gender were identified at dissection in the laboratory. Identification of the age of the individual was based on the level of atrophy of the thymus [41], as well as the body mass and status of the sex organs. We used three age categories: adults, subadults, and juveniles [42].

Out of 1450 small mammals trapped in the orchards, plantations, and meadows, 732 were granivores. This group was equally strongly dominated by *A. flavicollis* (374 individuals, 51.1%) and *A. agrarius* (346 individuals, 47.3%), while the *M. minutus* proportion was quite small (12 individuals, 1.6%).

The study was conducted in accordance with Lithuanian (the Republic of Lithuania Law on the Welfare and Protection of Animals No. XI-2271) and European legislation (Directive 2010/63/EU) on the protection of animals and was approved by the Animal Welfare Committee of the Nature Research Centre, protocol number GGT-7. In Lithuania, there is no need or obligation to obtain permission to snap trap small mammals. Trapping was unavoidable as we used the material for the pathogen analysis, collecting of organs (lungs, spleen, brain, etc.), and analysing of breeding parameters (from testes, uterus, and ovaries). In a few cases, trapped animals were killed by cervical dislocation.

### 2.3. Stable Isotope Analysis

We analysed the carbon and nitrogen stable isotopes in the hair of the trapped rodents. The most numerous species, *A. flavicollis* and *A. agrarius,* were sub-sampled in 2020. All suitable individuals of *M. minutus* were analysed. The distribution of the sampled individuals by species, habitat, age, and gender are presented in Table 2.

We used hair clipped from between the shoulders and stored them dry in separate bags for each individual. Dirty (covered by soil or blood) samples were washed in deionized water and methanol, and then dried. Very dirty samples were discarded.

Analyses were conducted at the Centre for Physical Sciences and Technology, Vilnius, Lithuania, using an elemental analyser (EA) (Flash EA1112) coupled to an isotope ratio mass spectrometer (IRMS) (Thermo Delta V Advantage) via a ConFlo III interface (EA-IRMS). The detailed analysis procedure and equipment used are described in [39]. Five percent of the samples were run in duplicates and the obtained results for these samples were averaged.

As reference materials, we used Caffeine IAEA-600 (*δ*^13^C = −27.771 ± 0.043‰, *δ*^15^N = 1 ± 0.2‰), Ammonium Sulfate IAEA-N-1 (*δ*^15^N = 0.4 ± 0.2‰), and Graphite USGS24 (*δ*^13^C = −16.049 ± 0.035‰) provided by the International Atomic Energy Agency (IAEA). These standards were run every 12 samples. Repeated analysis of these reference materials gave a standard deviation of less than 0.08‰ for carbon and 0.2‰ for nitrogen [39].

### 2.4. Statistical Analyses

Co-occurrence of the granivore species were analysed using the proportions of sites or trapping sessions in the habitats in which two or three granivorous species were co-trapped. A 95% confidence interval was calculated using the Wilson method, while the difference of proportions was evaluated using the chi-square method. As the number of sites in which any of the three species were absent was <10, the G-test for proportions (e.g., https://elem.com/~btilly/effective-ab-testing/g-test-calculator.html, accessed on 10 May 2021) was not used.

The *δ*^13^C and *δ*^15^N values in the samples were expressed as the arithmetic mean ± 1 SE and the range (min–max). Outliers were not excluded. The normalities of the distributions of the *δ*^15^N and *δ*^13^C values were tested using Kolmogorov–Smirnov’s D test. Based on the conformity to normal distribution of both isotope values in all species and in all habitats with sample size *n* ≥ 5, parametric tests were further applied.

The isotopic niches of the species were analysed using the parameters of TA (total area), SEA (standard ellipse area), and SEAc as corrected central ellipses, unbiased for the sample size [43]. The positions of species and intraspecific groups, including those with sample size *n* < 5, were visualized in isotopic biplots.

We used GLMM to find the influence of the species, habitat type (orchard, plantation, or meadow), habitat age, and intensity of agriculture as categorical factors on the dependent parameters: the *δ*^15^N and *δ*^13^C values. We also used two continuous predictors, namely year and season (summer or autumn), to control temporal data variability. Hotteling’s two sample T^2^ test for significance was used to test the significance of the model and eta-squared for the influence of the single factor. Intraspecific differences (between males and females and between age groups) were tested with parametric ANOVA using Wilk’s lambda test for significance. Differences between groups were evaluated with the post-hoc Tukey’s test and differences between pairs of variables with Student *t*-test.

The isotopic niches of the granivore species were calculated with SIBER [43], which we ran in R version 3.5.0 (https://cran.r-project.org/bin/windows/base/rdevel.html, accessed on 12 September 2020). Isotopic biplots were visualized using SigmaPlot version 12.5 (Systat Software Inc., San Jose, CA, USA). The proportions were evaluated in WinPepi version 11.39 software (Abramson, J., Jerusalem, Izrael). All other calculations were performed using Statistica for Windows version 6 (StatSoft, Inc., Tulsa, OK, USA).

## 3. Results

### 3.1. Sympatry and Syntopy of the Granivore Species

At 16 sites, *A. flavicollis* and *A. agrarius* were sympatric (trapped at the same site in the same habitat but not necessarily at the same time). All three granivorous species were sympatric in seven sites (all sites where *M. minutus* was present). Data on the sympatry of *A. flavicollis*, *A. agrarius*, and *M. minutus* are shown in Table 1.

Out of 18 investigation sites, *A. agrarius* was trapped in 17 (94%, CI = 74–99%) and *A. flavicollis* in 16 sites (89%, CI = 67–97%), the difference notably not significant. *M. minutus* was trapped in seven sites (39%, CI = 20–61%), thus the distribution of this species was significantly narrower than that of *A. agrarius* (χ^2^ = 9.48, *p* = 0.002) and *A. flavicollis* (χ^2^ = 12.15, *p* < 0.001). All seven sites with *M. minutus* were sympatrically inhabited by all three granivore species.

*A. flavicollis* was most frequently trapped in apple orchards (74% of trapping sessions), significantly exceeding the species’ presence in control meadows (39%, χ^2^ = 14.25, *p* < 0.001) and even more significantly than in other habitats, which themselves had no differences between them (Table 3). *A. agrarius* was present in all habitats including control ones with equal frequency. In *M. minutus*, the frequency of trapping in apple orchards, raspberry plantations, and control habitats was also equal (differences not significant).

Data on the syntopy of the granivorous species are presented in Table 3. The syntopy of the species’ pairs and triplets was the same in all habitats. The frequency of occurrence of syntopic *A. flavicollis* and *A. agrarius* depended on the presence of the last species, while syntopy with *M. minutus* was determined by its presence in the apple orchards, raspberry plantations, and control habitats.

### 3.2. Interspecific and Habitat-Based Differences of Isotopic Niche in Syntopic Granivores

Irrespective of other factors, the widest trophic niche according to *δ*^13^C was found in *A. agrarius*, significantly exceeding that of *A. flavicollis* (range of −17.41‰ vs. −13.46‰, respectively, t = 4.00, df = 456, *p* < 0.001). The range of *δ*^13^C (10.81‰) in *M. minutus* did not differ from either of the *Apodemus* species. The widest trophic niche according to *δ*^15^N was characteristic of *M. minutus* (range 17.62‰), significantly exceeding the range of *A. agrarius* (15.52‰, t = 3.94, df = 233, *p* < 0.001) and of *A. flavicollis* (11.45‰, t = 6.36, df = 243, *p* < 0.001). The *δ*^15^N range in *A. agrarius* was wider than in *A. flavicollis* (t = 10.24, df = 456, *p* < 0.001). Thus, species separation according to *δ*^15^N was full, while according to *δ*^13^C, it was only partial. The ranges and central positions of the stable isotope ratios of the granivores by habitat are presented in Appendix A.

The cumulative influence of species, habitat type, habitat age, intensity of agriculture, year, and season was significant for the distribution of both δ^13^C (F_11,256_ = 5.08, *p* < 0.001) and δ^15^N (F_11,256_ = 13.23, *p* < 0.001) values. While analysing univariate results, we did not find the influence of the year (Hotelling’s T^2^ = 0.001, *p* = 0.89). Other categorical factors were all significant: species (T^2^ = 0.44, *p* < 0.001, eta^2^ = 0.18), habitat type (T^2^ = 0.11, *p* < 0.001, eta^2^ = 0.05), intensity of agriculture (T^2^ = 0.09, *p* < 0.001, eta^2^ = 0.04), season (T^2^ = 0.04, *p* < 0.01, eta^2^ = 0.04), and habitat age (T^2^ = 0.04, *p* < 0.05, eta^2^ = 0.02).

The univariate analysis disclosed that season and habitat age influences were expressed only for the *δ*^15^N variation (F = 7.17, *p* < 0.01 and F = 3.51, *p* < 0.05, respectively), while the intensity of agricultural measures were expressed only for the *δ*^13^C variation (F = 8.17, *p* < 0.001). Mice species and habitat type, both being the strongest factors, influenced *δ*^13^C and *δ*^15^N variation (species: F = 14.39, *p* < 0.001 and F = 46.12, *p* < 0.001, respectively; habitat: F = 4.08, *p* < 0.01 and F = 4.53, *p* < 0.005, respectively). Therefore, further analysis of the distribution of the *δ*^13^C and *δ*^15^N values was conducted according to species and habitat (Figure 2).

Separation of species according to *δ*^13^C was not expressed in all habitats. Average values of *δ*^13^C in the hair of *A. agrarius* exceeded those of *A. flavicollis* in the apple orchards (Tukey’s HSD, *p* < 0.01, Figure 2b), plum orchards (*p* = 0.06, Figure 2c), and raspberry plantations (*p* < 0.005, Figure 2d). Despite the species being visually different from the other granivore species (Figure 2b), the average value of *δ*^13^C in *M. minutus* in the apple orchards was significantly smaller than only that in *A. flavicollis* (*p* < 0.02). In the control habitats, the variation of *δ*^13^C values in *M. minutus* was very wide (Figure 2f).

According to *δ*^15^N, granivorous species were separated irrespective of habitats (Figure 2a). The average values of *δ*^15^N in *A. agrarius* exceeded those of *A. flavicollis* in apple (Figure 2b, *p* < 0.001) and plum (Figure 2c, *p* < 0.10) orchards, in raspberry (Figure 2d, *p* < 0.002) and currant (Figure 2e, *p* < 0.01) plantations, and in control habitats (Figure 2f, *p* < 0.001). The average values of *δ*^15^N in *M. minutus* exceeded those in *A. agrarius* and *A. flavicollis* in the apple orchards (Figure 2b, *p* < 0.01 and *p* < 0.001, respectively). In the control habitats, the average values of *δ*^15^N in *M. minutus* exceeded those in *A. flavicollis* (Figure 2f, *p* = 0.015), while the difference from *A. agrarius* was not significant (*p* = 0.17).

The core dietary niches of the three granivorous species, shown as central ellipses in the isotopic space, were fully separated in the apple orchards and raspberry plantations (Figure 3a,b). A small overlap of 3.2% from the total area of the core niche of *M. minutus* was found with the core dietary niche of *A. agrarius* in control habitats (Figure 3c). When only two granivorous species were present, namely *A. flavicollis* and *A. agrarius*, the core dietary niches were separated in the plum (Appendix A) and currant orchards (Appendix A).

In the apple orchards, control habitats, and raspberry plantations, the total area of the trophic niche was widest in *A. agrarius*. The central niche in central habitats was widest in *M. minutus* (see also Figure 3c). In the plum orchards, the widest trophic niche was characteristic to *A. flavicollis* (Appendix A); please note, that area cannot be calculated in the small (*n* < 5) sample size (Appendix A).

### 3.3. Intraspecific Differences of the Isotopic Niche in Syntopic Granivores

Irrespective of habitat, intraspecific (age or sex-based) differences of the isotopic niche were not present in any granivorous species (Figure 4); thus, resources were equally used by males and females, adults, subadults, and juveniles of *A. flavicollis*, *A. agrarius*, and *M. minutus.*

## 4. Discussion

The assembly of the rodent community is inevitably based on the sharing of resources by the species. Resource-sharing is not easy to investigate; therefore, stable carbon and nitrogen isotopes in the tissues or hair of rodents may be used as a proxy, representing the diet of the individual and the trophic niche of the species [44]. Used at the level of animal communities, stable isotope analysis may show not only the dimensions of the trophic niche (e.g., niche width), but also the resource partitioning [9,23,45,46]. Therefore, as mentioned by Post, “stable isotope techniques combine benefits of both the trophic-level and food web paradigms in food web ecology” [47].

Separation of the isotopic niches has been shown in a community of African rodents: in the case of trophic niche overlap, spatial distribution was different [44]. This conforms to the hypothesis that species coexistence should be ensured by the partitioning of resources in space and time if they compete for this resource [25]. The hypothesis was also confirmed for rodents in Africa [44,48], for a flooded meadow in Lithuania [9], and for sympatric/syntopic voles of the genus *Microtus* in Lithuania [23]. The co-occurrence of species using similar resources has an influence on the trophic niche width and the partitioning of resources [36,37,38,46,48].

In the current study, we found that the isotopic niches (used as a proxy for trophic niches) of the three syntopic granivorous rodent species (*A. flavicollis*, *A. agrarius*, and *M. minutus*) in the commercial orchards, plantations, and control habitats (not subjected to agricultural activities used in orchards and plantations) were separated. The separation according to *δ*^15^N was better expressed than the separation according to *δ*^13^C. The core dietary niches of these species living syntopically were fully separated in the apple orchards and raspberry plantations, while there was an insignificant overlap of the core niche between *M. minutus* and *A. agrarius* in control habitats.

Two granivore species, namely *M. minutus* and *A. agrarius,* were not separated in flooded meadows in which resources were abundant for most of the small mammal species from the four guilds-granivores, omnivores, herbivores, and omnivores [9]. From the perspective of co-occurrence, the syntopy of granivores was less expressed than that in herbivore species in the commercial gardens and orchards [23,31]. Therefore, we should presume that these agricultural habitats are richer in resources for herbivores.

The role of the anthropogenic impact on the partitioning of resources in rodents has not been extensively studied, especially in the middle latitudes. For birds, it is known that deforestation in the tropic zone and the resulting loss of habitats influences bird diet and habitat use, with proof obtained by stable isotope analysis: the niches of frugivores, insectivores, nectarivores, and omnivores were narrower in human modified landscapes, while the niches of granivore birds were broader [49].

We agree with the view of Ribeira et al. [50] that more abundant resources result in broader isotopic niches (as a proxy for dietary niche). However, the presence of syntopic species of the same guild (granivores in the current study) seems to have a greater influence than resources. For example, the presence of the brown rat (*Rattus norvegicus*) affected the diet of the black rat (*Rattus rattus*) and the house mouse (*Mus domesticus*), shifting their temporal activities and narrowing trophic niches [51]. In agricultural areas in which resources for granivores are more limited than in flooded meadows, the isotopic niche was wider in *A. agrarius* (see Reference [9]). In syntopic herbivores from orchards and plantations, the range of *δ*^13^C values was not significantly greater than that in herbivores from flooded areas. However, the range of *δ*^15^N values was three to four times wider [9,23].

Depending on the habitat and latitude, seasonal changes in the isotopic niche may be expressed [44,50,52], not expressed [23], or absent (this paper).

Resource use may be strongly influenced by competition. In the case of sympatric Keen’s mice (*Peromyscus keeni*) and dusky shrews (*Sorex monticolus*), the first being herbivorous (https://animaldiversity.org/accounts/Peromyscus_keeni/, accessed on 5 June 2021) and the second insectivorous (https://animaldiversity.org/accounts/Sorex_monticolus/, accessed on 5 June 2021), their stable carbon and nitrogen values did not differ after a population crash in the mice [53]. Similar interspecific relations were also described among herbivores [54] and omnivorous *Peromyscus* species [30].

We found that the sympatric and syntopic co-occurrence of granivorous yellow-necked, striped field, and harvest mice in commercial orchards, berry plantations, and nearby meadows is more frequent than that of herbivorous *Microtus* voles [23]. Our findings show that the analysed agricultural habitats (fruit orchards and berry plantations) are special for syntopic rodent species, therefore requiring further investigation into the isotopic niche, resource partitioning, and influence of guild diversity (also, possibly, species abundances) on trophic ecology.

## 5. Conclusions

Syntopic living determines the separation of granivores in the isotopic space with full separation of species according to *δ*^15^N values, this presuming different amounts of food of animal origin in their diet.The separation of the core dietary niches in the apple orchards and raspberry plantations proved that these agricultural habitats are resource-scarce for granivores.Intraspecific differences of the isotopic niche were not present in any of the three granivorous species.

## Figures and Tables

**Figure 1 animals-11-02375-f001:**
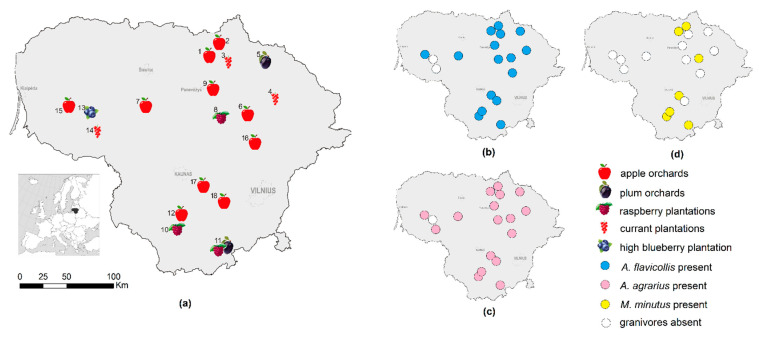
Commercial fruit orchards and berry plantations. (**a**) At the trapping sites in Lithuania, 2018–2020. The distribution of the granivorous species: (**b**) *A. flavicollis*, (**c**) *A. agrarius*, and (**d**) *M. minutus*.

**Figure 2 animals-11-02375-f002:**
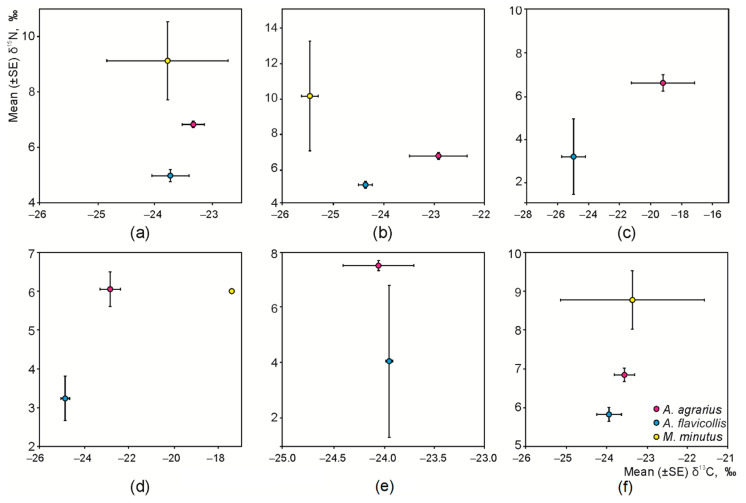
Distribution of syntopic granivore species according to the stable isotope ratios: (**a**) irrespective of habitat, (**b**) apple orchards, (**c**) plum orchards, (**d**) raspberry plantations, (**e**) currant plantations, and (**f**) meadows, as control habitats.

**Figure 3 animals-11-02375-f003:**
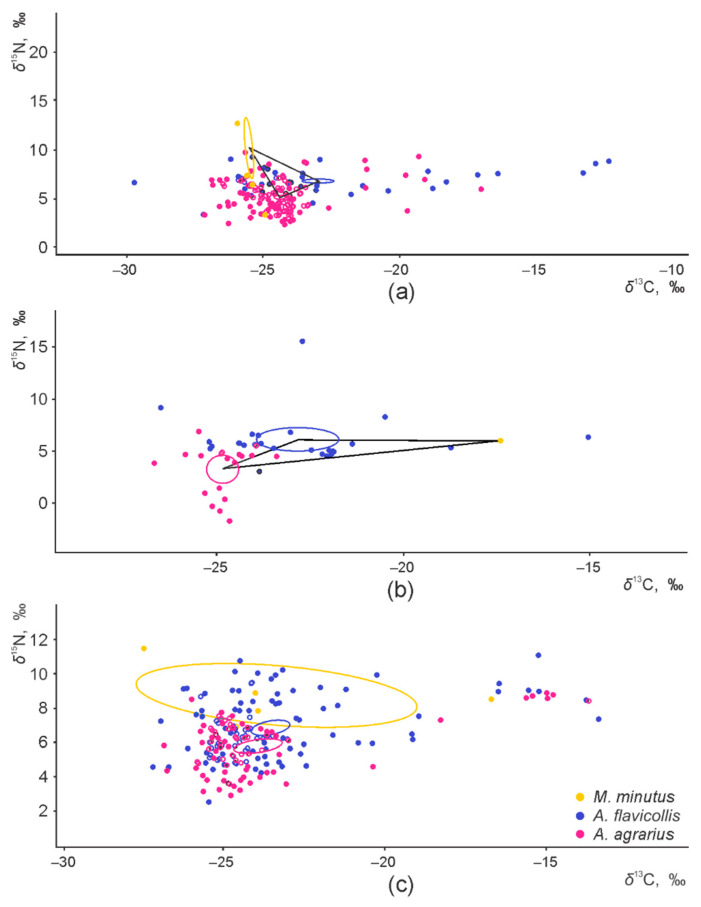
Central ellipses of the three syntopic granivorous species in the isotopic space, representing fundamental niches in apple orchards (**a**), raspberry plantations (**b**), and control habitats (**c**).

**Figure 4 animals-11-02375-f004:**
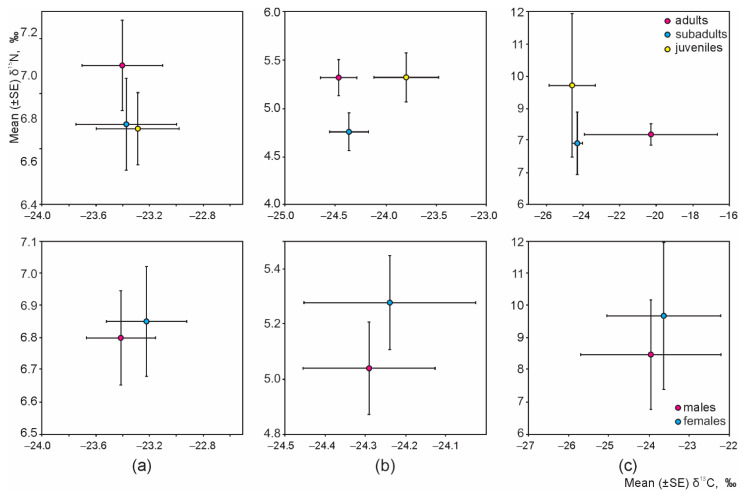
Intraspecific differences of the isotopic niches of *A. flavicollis* (**a**), *A. agrarius* (**b**), and *M. minutus* (**c**), irrespective of habitat.

**Table 1 animals-11-02375-t001:** Sympatry of granivorous rodents in fruit orchards, berry plantations, and control meadows of Lithuania, 2018–2020. Abbreviations: TE, trapping effort, trap days; TS, number of trapping sessions; NSM, number of all small mammals trapped; NH, number of habitats investigated; Af, *A. flavicollis*; Aa, *A. agrarius*; Mm, *M. minutus*; Af + Aa, by two species living syntopically; and All, by all three species living syntopically.

Habitat	TE	TS	NSM	NH	Number of Habitats Inhabited by				
					Af	Aa	Mm	Af + Aa	All
Apple orchards	11,268	46	567	10	10	8	5	8	4
Plum orchards	600	8	26	2	2	1	0	1	0
Raspberry plantations	1450	15	76	3	3	3	1	2	1
Currant plantations	2950	14	180	3	2	2	0	2	0
Control habitats	8260	82	532	17	11	13	3	9	3

**Table 2 animals-11-02375-t002:** Sample sizes by habitat of the three species of granivorous rodents used for stable isotope analysis. Abbreviations: N, number of analysed individuals; m/f, numbers of males and females; and a/s/j, numbers of adults, subadults, and juveniles, respectively.

Habitat	*Apodemus flavicollis*			*Apodemus agrarius*			*Micromys minutus*		
	N	m/f	a/s/j	N	m/f	a/s/j	N	m/f	a/s/j
Apple orchards	127	67/60	56/44/27	44	26/18	11/15/18	5	2/3	0/0/5
Plum orchards	4	1/3	1/0/3	5	3/2	0/0/5	0		
Raspberry plantations	19	13/6	3/10/6	26	16/10	9/2/15	1	1/0	0/0/1
Currant plantations	2	2/0	1/1/0	35	16/19	9/9/17			
Control habitats	82	41/41	21/26/35	114	62/52	26/34/54	5	2/3	2/2/1

**Table 3 animals-11-02375-t003:** Syntopy of granivorous rodents in commercial orchards, berry plantations, and control habitats. 2018–2020. Abbreviations: TS, number of trapping sessions; NAf, number of habitats inhabited by *A. flavicollis*; NAa, by *A. agrarius*; NMm, by M. *minutus*; NAf + Aa, AAf + Mm, and NAa + Mm, by two species living syntopically; and NAll, by all three species living syntopically.

Habitat	TS	NAf	NAa	NMm	NAf + Aa	NAf + Mm	NAa + Mm	NAll
Apple orchards	46	34	14	5	13	4	4	4
Plum orchards	8	3	1		1			
Raspberry plantations	15	5	5	1	3	1	1	1
Currant plantations	14	3	4		3			
Control habitats	82	32	26	5	14	4	5	4

## Data Availability

After publication, research data will be available from the corresponding author upon request. The data are not publicly available due to usage in the ongoing study.

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
