# Peer review of "Isotopic Niche of Syntopic Granivores in Commercial Orchards and Meadows"

_animals, 2021, doi:10.3390/ani11082375_

Round 1
Reviewer 1 Report
Review (Animals [ISSN 2076-2615], Isotopic Niche of Syntopic Granivores in Commercial Orchards and Meadows by Linas Balčiauskas et al.
Although generally well written and well organized, this paper would benefit from review of an individual whose first language is English.
Abstract
Both here and in the Simple Summary, the openings are a bit abrupt. It would help to hypothesize why trophic niches may vary between agricultural habitats and meadows and why this may be important.
Line 6 of abstract change "interspecies" to "interspecific"
Last statement in Abstract is vague and suggests little could be concluded. Be more specific about findings.
Introduction
I suggest rewording this "A. agrarius and A. flavicollis are subdominants, dominating in a number of cases [23]." I suggest: "A. agrarius and A. flavicollis are subdominants, but occasionally dominate in some cases [23]."
This statement is not clear. "As main hypothesis, we expected differences in their diet (expressed, as a proxy, by the isotopic space of the species), minimizing interspecies competition for food resources." This statement is not clear. Are you expecting divergence of their diet because of their convergence on agricultural habitats?
Methods
The methods are generally sound although language could be improved some.
Results
Table 1 is difficult to read and needs some reorganization. TS is not defined.
Figure 2. I assume the last three figs should be labelled d, e, and f instead of d, d, and d.
Discussion
The Discussion is generally well done.
Author Response
Dear Rev#1
thank you for your comments. We accepted all of these, and please see all answers, point by point, in the attachment.

Reviewer 2 Report
Thank you very much for an opportunity to revise the manuscript. It was a very interesting reading. a lot of nice and inspiring results. Yet, some in points this improved by addition of clear explanations, please my comments below.
lines 16-18: Under the domination of striped field 16 and yellow-necked mice, syntopic co-occurrence depended on the presence of harvest mouse in the 17 apple orchards, raspberry plantations and meadows that served as control habitats: I am sorry, but for me this sentence is unclear, especially it is in the simple summary, which should be easy to understand for non-specialists. I guess there should be some explanation what δ 15N and δ 13C will show, i.e. will they reflect different food components?
lines 23-24, and lines 37-38: Based on these 23 findings, further investigation into the influence of guild diversity on trophic ecology in agricultural 24 habitats is required: I am not convinced by this. Why is it needed? Such sentence should not be the main conclusion from the article, please refer to your goal here.
lines 27-30: We analysed the isotopic niches of striped field (Apodemus 27 agrarius), yellow-necked (Apodemus flavicollis) and harvest (Micromys minutus) mice living sympat- 28 rically and syntopically in commercial orchards and berry plantations, as well as nearby meadows 29 that were used as control habitats.: Why meadows were control habitats? According to the title orchards were compared to the meadows. What about berry plantations? Were they treated as orchards? It is said that you looked at differences in the isotopic niches of these three species but did simply compare orchards and meadows as two habitat types? Or did you expect higher/lower competition in orchards vs meadows?
keywords: they mostly repeat those from the title. Also, Lithuania does not seem to be the best keyword. How about terms like interspecific competition, intraspecific competition? maybe δ 15N, δ 13C?
lines 87-90: but did you expect this competition to be different in transformed vs natural habitats? Is this why meadows served as control habitats?
Figure 1: I got lost a bit, I can see al lot of apple orchards and this is clear but just one blueberry plantation (and this was mentioned in the abstract) and some plum and currant plantations (and they were not mentioned at all). Are all of those named commercial orchards? (I mean including currant and plums but excluding berry plantations?), as this is what the caption suggests.
And, but this is just my feeling, I would make apples a bit smaller as they greatly dominate over other symbols. What do white dots show? Absence of any species? This should be stated.
line 110: IN three of these…
lines 118-125: I feel this is part of the results and not the methods
Table 1-3: I got a bit confused, there are no blue berry plantations listed in the tables, Ok, I looked at the figure 1 and saw that no rodent were trapped at a blue berry plantation. But I think this needs to be said somewhere in the results and, what is important, you should clearly say what you mean by orchards and berry plantations (listing those habitats in the bracket).
Figure 2: d is repeated three times, should be e, f I suppose. Apart from this, I like the results figures, they are very clear and informative.
line 298: better we found not we find
lines 310-311: Therefore, we should presume that these agricultural habitats are richer in resources for herbivores: you mean meadows here? By the way, this reference to the transformation level of habitats is sth I miss in the abstract.
line 337: when you say ‘agricultural habitats’ you mean meadows as well?
lines 342-343: is more frequent than that of herbivorous Microtus voles.: but this study did not test Microtus species…
lines 347-349: this repeated results, not conclusion, I miss real conclusions here, which will point to the relevance of the obtained results
Author Response
Dear Rev#2
thank you for your comments. We accepted all of these, and please see all answers, point by point, in the attachment.
